ezRAD: a simplified method for genomic genotyping in non-model organisms

Toonen Robert J. 1 toonen@hawaii.edu
Puritz Jonathan B. 1
Forsman Zac H. 1
Whitney Jonathan L. 1
Fernandez-Silva Iria 1
Andrews Kimberly R. 1
Bird Christopher E. 2
1 Hawaiʻi Institute of Marine Biology, School of Ocean & Earth Sciences & Technology, University of Hawaiʻi at Mānoa , Coconut Island, Kāneʻohe, HI , United States
2 Department of Life Sciences, Texas A&M University - Corpus Christi , Corpus Christi, TX , United States
Medina Mónica
Electronic publication date: 2013 Nov 19
Publication date: 2013
Volume: 1
Electronic Location ID: e203
Received 2013 Jul 16; Accepted 2013 Oct 13
Copyright: © 2013 Toonen et al.
Copyright year: 2013
Copyright holder: Toonen et al.
License: This is an open access article distributed under the terms of the Creative Commons Attribution License, which permits unrestricted use, distribution, and reproduction in any medium, provided the original author and source are credited.
License URL: https://creativecommons.org/licenses/by/3.0/

Keywords: RAD tag, RADseq, RAD-seq, Restriction site associated DNA (RAD), Next-generation sequencing, NGS, Genotype-by-sequencing

Funding: NSF OCE-1260169 NOAA NMSP MOA#2005-008/66882 This work was funded in part by the National Science Foundation (Grant OCE-1260169), and the National Oceanic and Atmospheric Administration (NMSP MOA#2005-008/66882). Additional support for the co-authors came from the National SeaGrant Program, the Hawaiʻi Coral Reef Initiative, the Seaver Institute, Fulbright – Spanish Ministry of Science and Technology, and the Japanese Society for the Promotion of Science. The funders had no role in study design, data collection and analysis, decision to publish, or preparation of the manuscript.

==============================
Here, we introduce ezRAD, a novel strategy for restriction site–associated DNA (RAD) that requires little technical expertise or investment in laboratory equipment, and demonstrate its utility for ten non-model organisms across a wide taxonomic range. ezRAD differs from other RAD methods primarily through its use of standard Illumina TruSeq library preparation kits, which makes it possible for any laboratory to send out to a commercial genomic core facility for library preparation and next-generation sequencing with virtually no additional investment beyond the cost of the service itself. This simplification opens RADseq to any lab with the ability to extract DNA and perform a restriction digest. ezRAD also differs from others in its flexibility to use any restriction enzyme (or combination of enzymes) that cuts frequently enough to generate fragments of the desired size range, without requiring the purchase of separate adapters for each enzyme or a sonication step, which can further decrease the cost involved in choosing optimal enzymes for particular species and research questions. We apply this method across a wide taxonomic diversity of non-model organisms to demonstrate the utility and flexibility of our approach. The simplicity of ezRAD makes it particularly useful for the discovery of single nucleotide polymorphisms and targeted amplicon sequencing in natural populations of non-model organisms that have been historically understudied because of lack of genomic information.

Introduction

Next-generation sequencing (NGS) has provided unprecedented access to genomic information at ever-increasing speed and reduced cost (Mardis, 2008). Until recently, profiling a large number of loci was only realistically possible for organisms with well-developed genomic resources, and the high cost of developing these resources has been a major impediment to studies for non-model organisms. Despite the rapid advances of sequencing technology, and dramatic reduction in cost associated with those advances, whole-genome sequencing remains a costly hurdle to undertake for marker development in non-model organisms, especially for species with large genomes. In the fields of phylogeography, phylogenetics, and population genetics, the majority of studies do not require whole-genome sequencing, but rather a spread of loci across the genome. As a result, there has been considerable interest in simple and more cost-effective approaches to using reduced representation genome sequencing, such as restriction site associated DNA sequencing, or RADseq. RADseq effectively reduces genome complexity and size by resequencing only stretches of genomic DNA adjacent to restriction endonuclease sites, providing high coverage of homologous portions of the genome from multiple individuals for comparatively low cost and effort. A multitude of strategies have emerged for RAD sequencing, including the original method (Baird et al., 2008; Etter et al., 2011; Hohenlohe et al., 2010), genotype-by-sequencing, or GBS (Elshire et al., 2011; Sonah et al., 2013), 2-enzyme GBS (Poland et al., 2012), 2b-RAD (Wang et al., 2012), and ddRAD (Peterson et al., 2012). The RADseq approach provides a powerful tool for a wide range of genetic studies and is rapidly changing the field as a result (reviewed by Rowe, Renaut & Guggisberg, 2011).

Despite rapidly gaining popularity, the application of RADseq has been limited to primarily model or emerging model species (Baird et al., 2008; Chutimanitsakun et al., 2011; Emerson et al., 2010; Hohenlohe et al., 2010), with only a single marine invertebrate species RADseq dataset (for the model organism Nematostella vectensis) published to date (Reitzel et al., 2013). Although it is debatable what the underlying cause of this delayed application is, applying existing protocols to non-model marine invertebrates can be challenging due to a variety of unknowns such as genome size and frequency of restriction sites. Additionally, many of the existing protocols require a significant initial investment for labs focused on Sanger sequencing and microsatellite typing. Thus, we sought to develop a simplified and general approach to RADseq that requires little to no optimization and would enable access to this powerful new approach among taxonomic groups across the tree of life. Here, we outline a novel RAD strategy that uses any restriction enzyme (or combination of enzymes) which cuts frequently enough to produce fragments suitable for sequencing, and then uses the standard Illumina TruSeq library preparation with agarose gel (or SPRI-bead) size-selection to target the fragments to be sequenced. The approach is flexible and scalable, making it possible for virtually any lab to send out restriction endonuclease digested DNA for RAD sequencing with no additional investment beyond the cost of the core lab costs for library preparation and sequencing itself. This simplification opens the door to RAD sequencing for any lab with the ability to extract DNA and perform a restriction digest, a very low technical bar for the application of NGS. Furthermore, this method is compatible with a wide range of restriction enzymes, and does not require the purchase of new adapters for each new enzyme. Therefore, ezRAD provides flexibility for optimizing the number of unique fragments to be sequenced by simple modifications to the restriction enzyme and/or size selection range used. Here, we report and apply ezRAD across a wide taxonomic diversity of non-model organisms to demonstrate the utility of this approach. This generalized approach, using the standard Illumina TruSeq library preparation kit, will allow researchers to apply RADseq technology to a wide array of research questions.

Methods

DNA extraction and quantification

High molecular weight DNA was extracted from preserved tissue samples using a variety of methods. For Patiria miniata, Porites compressa, Porites lobata,and Stenella longirostris, the E.Z.N.A. MicroElute Genomic DNA extraction kit (Omega) was used according to the manufacturer protocol. DNA from Cryptasterina hystera, Cryptasterina pentagona, Pocillopora damicornis, and Cellana talcosa was extracted using the DNeasy tissue extraction kit (Qiagen), and DNA from Paracirrhites arcatus was extracted using a standard Phenol-Chloroform procedure with the addition of RNase. All extractions were inspected on 2% agarose gels for the presence of impurities and lower molecular weight DNA. Samples with “smear” gel patterns were subsequently purified with AmpureXP (Agentcourt) SPRI beads using a 2:1 template to bead volume ratio. Subsequently, all extractions were quantified using AccuBlue High Sensitivity fluorescence assay on a SpectraMax M2 plate reader, using the standard protocol for a 96 well assay (Application Note #22) with the adjustment of using the AccuBlue dye excitation and emission spectra. For all libraries, 1.5 µg of DNA was precipitated in 1/10 volume Sodium Acetate and two volumes 100% ethanol at −70°C for 30 min. DNA was pelleted by 15 min of centrifugation at 12,000 rpm. Pellets were resuspended in 24 µl of dI water with 0.1 M Tris at 65°C.

Digestion

DNA was digested simultaneously with the isoschizomers MboI and Sau3AI (NEB) to minimize any potential impacts of methylation of DNA in digesting the genomic libraries. Each digestion was performed in 50 µl reactions: 5 µl NEB Buffer 4, 0.5 µl BSA, 2 µl MboI, 2.5 µl Sau3AI, 18 µl of DNA template (roughly 1.125 µg) from above, and 22 µl of dI water. Digestions were incubated at 37°C for 3–6 h and then cleaned using 80 µl of AmpureXP beads per reaction and eluted in 20 µl of water.

Illumina library preparation

Cleaned digestions were inserted directly into the Illumina TruSeq DNA kit following the Sample Preparation v2 Guide starting with the “Perform End Repair” step. Digestions can be inserted into any of the three available Illumina TruSeq DNA kits including the newest PCR-free and Nano kits. Due to the challenges of working with non-model marine invertebrates, many of our libraries had less than 1 µg of high molecular weight DNA with which to start the library preparation, but the Nano kit was not yet available. Due to the low starting concentration, we performed the PCR enrichment before gel extraction, but have had better success with the Nano kit since its release, and would recommend that approach for low initial DNA concentrations. We generally followed the TruSeq protocol, but attempted to save reagents and further lower costs by performing nearly all reaction steps in 1/3 of the recommended volumes (see detailed protocol - File S1), although such modifications are not necessary for the protocol. In brief, digested libraries were end repaired, 3′ ends were adenylated and TruSeq adapters were ligated to the digested genomic DNA sample. Libraries were then size-selected following the Illumina TruSeq protocol using a 2% low-melt agarose gel with 1X TAE buffer run at 120 V for 120 min. The 400–500 bp fragments (of which ∼120 bp are the ligated adapters) were cut out with a sterile scalpel blade for each individual sample and DNA was recovered using the Qiagen MinElute Gel Extraction Kit following manufacturer instructions. After gel extraction, libraries were validated by visualization on an Agilent 2100 BioAnalyzer, quantified using qPCR, and pooled (performed by the Hawaiʻi Institute of Marine Biology EPSCoR Core sequencing facility). Pooled libraries were then sequenced as paired-end 100 bp runs on the Illumina GAIIx at HIMB.

Bioinformatics

The HIMB Core facility runs a standard quality control filter and parses the Illumina reads into fastq files sorted by index. Beyond that, a custom bash script (File S2) was used to automate read quality filtering, reference contig assembly, read mapping, SNP calling, and SNP filtering. A brief description of each step of the analyses follows below:

Raw FASTQ files were trimmed using the program Trim Galore! (http://www.bioinformatics.babraham.ac.uk/projects/trim_galore/) into two different read sets. The first set of reads had only adapter sequences removed and were subsequently saved for contig assembly. The second set of reads was trimmed for adapter sequences and also removed any base that had a quality score of less than 10 (90% probability of being correct). These reads were saved for mapping.

Rainbow (Chong, Ruan & Wu, 2012) clustered and assembled the first set of parsed fastq files into a final assembly of reference contigs. Rainbow is specifically designed to assemble contigs from RAD sequencing. In short, it first clusters reads together that are less than 4 bp apart. These clustered reads are then recursively divided into groups representing individual allele sequences. Individual allele sequences are then assembled and merged into a final set of RAD contigs.

Quality trimmed reads were then mapped to the reference contigs using BWA (Li & Durbin, 2009) with the MEM algorithm and default parameters (with the exception of altering the number of computational threads and restricting the output to only map scores of 10 and higher). SAM files were converted to BAM files using SAMtools (Li et al., 2009) and output was further restricted to reads with mapping quality above 15. BAM files were then merged and realigned around INDEL regions using the mpileup command of SAMtools with default parameters and the additional command of outputting per-sample read depths.

SNP calling was performed using VarScan2 (Koboldt et al., 2009; Koboldt et al., 2012) using the mpileup2snp command with default settings. The strand filter was removed because overlapping forward and reverse reads are not expected for the insert size of this library, and the minimum variant frequency was raised from 1% to 10%, meaning that within one library the minimum allele frequency had to be above 10% to be called a SNP. Finally, the p-value for a significant variant was raised from 0.01 to 0.05. Genotypes failing any of these filters are reported as missing.

The raw SNP calls were then filtered using two instances of VCFtools (Danecek et al., 2011). The first instance filters out INDEL loci, sites that were fixed for the minor allele, and SNPs that were not genotyped in 99% of the samples. The second round of filtering removes sites with less than 10× coverage and outputs the final set of SNP genotypes in VCF format.

Reads from the four Patiria miniata libraries were used for validation of the ezRAD technique and bioinformatics pipeline. For the first test, contigs generated by Rainbow (Chong, Ruan & Wu, 2012) for the ezRAD P. miniata libraries were aligned using BWA (Li & Durbin, 2009) with the MEM algorithm and default parameters to previously published P. miniata genomic contigs from GenBank (Assembly GCA_000285935.1). As a second test, genomic contigs were substituted for the ezRAD contigs in the analysis pipeline to directly compare the number of SNPs generated with different reference contigs from P. miniata.

Results

Taxonomic representation

We tested our generic protocol with no attempt at optimization across a range of taxonomic diversity including a marine mammal (Stenella longirostris), a coral reef fish (Paracirrhites arcatus), three echinoderms (Patiria miniata, Cryptasterina hystera & C. pentagona), a mollusk (Cellana talcosa), and three scleractinian corals (Porites compressa, P. lobata & Pocillopora damicornis). Across the range of metazoan diversity from cnidarians to vertebrates, the technique worked relatively well with no modifications or attempts at optimization (Table 1).

Table 1 Summary of ezRAD results from 2 lanes of Illumina GAIIx sequencing across a range of taxonomic diversity.

Organisms run with ezRAD include: the limpet Cellana talcosa; sea stars Cryptasterina hystera, C. pentagona & Patiria miniata; reef fish Paracirrhites arcatus; corals Porites lobata, P. compressa & Pocillopora damicornis; and the spinner dolphin Stenella longirostris. Lane use indicates the proportion of a single lane of PE100bp sequencing on the GAIIx flow cell. Library prep specifies what those libraries contained. Paired reads are the number of reads in each index parsed file returned after initial quality control (QC) filter from the sequencer. Reads and % Pass QC are the number of sequence reads remaining after excluding all sequences for which Phred scores were <20, contained adapter sequences, or were less than 20 bp long after adapters were cut. Mapped reads and High quality mapped reads are the number of reads overall and the number reads that passed quality control, respectively, that were assembled de novo into contigs. Contig statistics and polymorphic SNP counts reported here come from the bash script pipeline described in File S2.

Species	Lane
use	Library
prep	Paired
reads	Reads
passing
QC	% Pass
QC	No.
contigs	Mapped
reads	High
quality
mapped	Variable
sites	Shared
SNPs	>10X
SNPs
shared	>30X
SNPs
shared	SNPs
Per
contig	
Cellana
talcosa	1/3	2 pools of
24 individ	8,109,327	4,472,365	55.15	189,444	8,389,669	3,483,833	127,609	73,014	49,761	9,997	0.26	
Cryptasterina
hystera	1/12	1 pool of
10 individ	6,441,832	3,880,080	60.23	27,007	2,242,589	655,509	36,666	36,666	31,127	12,827	1.15	
Cryptasterina
pentagona	1/12	1 pool of
10 individ	1,804,165	325,137	18.02	4,354	1,452,171	148,580	9,501	9,501	8,058	3,538	1.85	
Patiria
miniata	3/4	4 pools of
20 individ	15,632,982	13,132,267	84.00	635,376	47,718,931	14,987,372	1,167,981	187,597	143,254	21,914	0.23	
Paracirrhites
arcatus	2/3	8 tagged
individ	13,586,625	5,261,386	38.72	205,360	22,892,817	3,088,806	171,712	2,705	2,447	366	0.01	
Paracirrhites
arcatus	1/6	2 pools of
4 individ	2,733,965	995,543	36.41	13,340	2,980,976	324,331	20,512	10,221	7,249	2,082	0.54	
Pocillopora
damicornis	1/6	2 tagged
individ	3,289,007	2,929,932	89.08	164,553	5,710,298	2,956,701	232,734	110,454	80,221	21,658	0.49	
Porites
compressa	1/4	3 tagged
individ	5,512,212	3,852,694	69.89	123,874	7,011,801	2,338,160	297,648	77,346	60,149	13,040	0.49	
Porites
lobata	1/4	3 tagged
individ	7,131,427	3,926,250	55.06	95,887	7,587,534	2,689,310	275,769	65,731	47,617	9,419	0.50	
Stenella
longirostris	1/3	4 pools of
2 individ	7,660,053	2,563,007	33.46	43,427	9,910,871	1,022,661	69,208	7,502	7,280	3,828	0.17	

Sequencing results

All attempted libraries yielded thousands to tens of thousands of variable base calls. Results varied by taxon (Table 1), but as with other RAD protocols, the two factors most directly linked to the number of reads per library passing quality control were initial DNA fragment sizes and overall genome size, as opposed to taxonomic relatedness. When holding the size selection range constant, large genome sizes and low molecular weight DNA resulted in increased numbers of non-homologous DNA fragments with lesser coverage. For example, Acropora digitifera has ∼420 megabase genome (Shinzato et al., 2011), but in combination with the suite of dinoflagellate, prokaryotic and eukaryotic symbionts inextricably associated with the coral holobiont (reviewed by Ainsworth, Thurber & Gates, 2010) this makes for an exceedingly large genomic pool from which to draw fragments for reduced representation genomic sequencing. Genome size is estimated from the amount of DNA (in picograms) contained in a haploid nucleus, taken from the Animal Genome Size Database (http://www.genomesize.com). The corals (1115 Mb for Siderastrea stellata, plus 1467–4694 Mb for Symbiodinium symbionts) have relatively lower coverage across each contig in comparison to species with smaller genome sizes, such as the limpets (421 Mb for Lottia gigantea) or the sea stars (743 Mb for Patiria miniata). The number of both fragments and putative SNPs identified for P. miniata may appear high, but the species is known to be extremely polymorphic — even by comparison to other sea stars in the Asterinidae (Keever et al., 2009; McGovern et al., 2010; Puritz & Toonen, 2011). Further, we validate these variable bases as putative SNPs against published genomic contigs (see ezRAD Validation below).

Samples with low molecular weight DNA extractions and large genome size produced the lowest quality among all libraries in our tests. Further, genomic DNA samples characterized by low molecular weight fragments were also characterized by reduced adapter ligation efficiency that led to a large number of sequenced fragments consisting of only adapter dimers with low quality scores. For the sea star, C. pentagona is a relatively large genome and the lowest molecular weight DNA in this study resulted in the lowest quality library (Table 1). However, despite only 18.02% of the sequence reads passing QC, we still discovered over 8,000 variable base sites which is more than sufficient for SNP discovery applications. Likewise, the fact that the arc-eye hawkfish (Paracirrhites arcatus) has a relatively large genome (714 Mb for Cirrhitichthys aureus) together with carryover of degraded DNA from the initial extraction, resulted in P. arcatus showing a relatively low percentage of reads that passed QC (Table 1).

ezRAD validation with published reference genomic contigs

ezRAD derived genomic sequencing reads of P. miniata were mapped to reference consensus sequences generated from the publicly available genomic contig data from P. miniata on GenBank. Overall, 532,467 of the 635,376 P. miniata contigs generated from the ezRAD analysis pipeline mapped with high quality to publicly available genomic contigs (MAPQ mean = 38.66; median 52.00; standard deviation 23.22). Approximately 15 million ezRAD reads mapped with high quality to the reference, versus ∼21 million reads from the publicly available data set (Table 2). Most importantly, the number of variable sites, shared SNPs, and quality-controlled SNP datasets were similar between the reference contigs from each of the two approaches (Table 2).

Table 2 Validation of ezRAD data against genomic contigs.

Comparison of ezRAD results using two different sets of reference contigs, the original ezRAD analysis pipeline contigs and published genomic contigs for the seastar Patiria miniata.

Reference type	ezRAD	Genomic contigs	
Number of contigs	635,376	179,756	
Mapped reads	47,718,931	26,130,869	
High quality mapped	14,987,372	21,997,385	
Variable sites	1,167,981	1,156,633	
Shared SNPs	187,597	151,742	
>10× Shared SNPs	143,254	114,620	

SNP discovery using pooled and unpooled libraries

We compare results obtained by making a single library per individual for each of eight individuals of the reef fish P. arcatus relative to two pools of four individuals each (Table 1). After normalizing for lane use, we generated 2.4× more high quality mapped reads for the eight individual libraries (4.6 × 106/lane) than for two pooled samples containing eight individuals (1.9 × 106/lane) and 2× more variable base calls (2.6 × 105/lane vs. 1.2 × 105/lane). When accounting for cost however, the sample prep and sequencing was 4× more expensive for the 8 individuals versus two pools of four individuals, making the pooling strategy more cost effective per variable base identified. Furthermore, because of some variability among individual libraries, we identified 11.9× more shared SNPs with >10× coverage between the pooled libraries (4.35 × 104/lane) than among the eight individual libraries (3.67 × 103/lane). For example, 975 shared SNPs were genotyped (at >10× mean coverage) in all 8 individual libraries, and 635 of those (65%) were also genotyped in both pooled libraries. By comparison, of the 3344 SNPs that were genotyped (at >10× mean coverage) between the 2 pooled libraries, only 626 (19%) of those were also genotyped in all 8 individual libraries. Even without normalization for lane use, we identified more shared SNPs of higher quality for lower cost from the two pooled libraries relative to the individual libraries (Table 1; File S3).

Discussion

ezRAD, a novel approach to reduced representation genomic sequencing, differs from existing RADseq methods primarily in that it requires very little technical expertise or laboratory equipment to complete. These benefits are achieved through the use of the Illumina TruSeq library preparation kits, which also makes it possible to send digested DNA to any core lab that offers library preparation as part of their Illumina service package. This method now makes RADseq possible for any lab with the ability to perform DNA extraction and restriction digestion, an extremely low technical expertise and equipment bar to achieve NGS capability.

ezRAD is similar in concept to several other recently developed RAD methods, such as GBS, 2-enzyme GBS, ddRAD and 2b-RAD (Elshire et al., 2011; Peterson et al., 2012; Poland et al., 2012; Sonah et al., 2013; Wang et al., 2012) in that we use frequent-cutting enzymes to generate fragments of the appropriate length for sequencing (usually between 300–500 bp), rather than using a sonication step to sheer DNA after digestion as in the original RADseq protocol (see Table 3 for a comparison among methods). ezRAD and ddRAD both use a size selection step to eliminate inappropriately-sized fragments generated by the restriction digest. In contrast, GBS relies on a PCR step to preferentially amplify shorter fragments over longer fragments; and 2b-RAD uses a special type of restriction enzyme (IIB enzymes) that cuts DNA into small, uniformly sized fragments (33–36 bp) suitable for sequencing. As in GBS, ddRAD & 2b-RAD, the number of unique fragments generated by ezRAD can be optimized by altering the restriction enzyme(s) used based on the frequency of cut sites in the genome (if known). ddRAD & ezRAD share the additional advantage that the number of fragments sequenced can be modified through the size selection step in the library preparation. However, ezRAD offers two advantages that simplify the process of choosing an appropriate restriction enzyme for a given organism and research question: (1) ezRAD generally targets just one restriction site (here both MboI and Sau3AI target GATC, but with different sensitivity to methylation). Assuming some knowledge of genome size and GC content, using a single cut site simplifies the calculations to predict the number of unique genomic regions within a given size range that will be generated through the digest. In practice any restriction enzyme, or combination of enzymes, that result in appropriately sized fragments could be used. (2) The adapters are not custom-designed for the enzyme(s) used, thereby allowing researchers to try many different restriction enzymes (or combinations of enzymes) without the costly investment of new adapters for each enzyme. The ability to quickly try multiple different enzymes may be particularly beneficial for recalcitrant genomes where no prior knowledge of genome content is available, as is the case in many non-model organisms. Regardless of such differences among techniques, for many applications optimization of restriction enzyme and size selection range are likely irrelevant for SNP discovery, because even without any attempt at optimization, and with as few as 18% of reads passing QC, we nevertheless discovered thousands of putative SNPs in each library here (Table 1).

Table 3 Comparison of most commonly used RAD sequencing methodologies and associated costs.

	No. of
enzymes	Cut
frequency	Shearing
required	Size
selection	Library prep
time &
required
expertise	Initial
outlay cost	Subsequent
library cost
per sample	Scalability to
reduce
overall cost
per sample	
ezRAD	1 or more	Frequent	No	Yes	Low	Very Low	Moderate	Low	
RAD tags	1	Rare	Yes	Yes	High	High	Low	Low	
GBS	1	Rare or
frequent	No	No	Moderate	High	Moderate to
very low	Low	
2-enzyme
GBS	2	Rare +
frequent	No	No	Moderate	High	Moderate to
very low	Low	
ddRAD	2	Frequent	No	Yes	Moderate	High	Very low	Moderate	
2b-RAD	1	Frequent	No	No	Moderate	High	Low	Moderate	

In order to highlight the flexibility and broad applicability of this approach, we provide an example application of this approach in which we prepared 30 libraries across a wide taxonomic range. With no adjustments to the protocol, and no attempt at optimization of any step to accommodate the taxonomic or genomic differences among the taxa, we successfully RAD sequenced a marine mammal, a fish, a mollusk, several echinoderms, and scleractinian corals. While the protocol obviously did not work equally well on all samples in the run, sequencing of all libraries yielded sufficient data for most applications from every taxon attempted (Table 1). Rather than a taxonomic bias in the success of the technique, the success of a library appears to be a direct result of the initial quality of DNA that went into the library preparation and the genome size. We find that starting with the highest possible molecular weight DNA and testing for the standard QC along the way makes the greatest difference in the amount of useable sequence data resulting from a run.

Costs

The total cost of preparing and sequencing our 30 libraries in two lanes of an Illumina GAIIx flow cell was $9,600. We were able to reduce the cost per dataset to ∼$320 (start to finish) by buying all reagents and constructing the libraries in our laboratory and sequencing through our HIMB core facility, making the library preparation cost on the order of ∼$60. Current prices for library preparation at academic institutions that offer the service in the USA are on the order of $200–300 for the TruSeq library preparation and approximately the same for 1/6th of MiSeq or 1/12th an Illumina GAIIx sequencing lane, making the total price of an ezRAD run on the order of $500–$600 USD if sent out. Costs could be further reduced on a HiSeq, but in this case ezRAD samples need to be run among lanes filled with other libraries to account for the fact that the first 4 bases will be the same (GATC cut site) on most sequences. Some will note that this cost is substantially higher than the published library prep costs for other RAD methods which can be as low as $5/sample, but such estimates do not include the cost of sequencing, which is the majority of the cost of RADseq. However, it is also important to note that there is a trade-off between the ultimate cost per library and the initial investment to begin the process of library development. Initial investment for most RADseq methods would include, at the minimum, acquiring a Solid Phase Reversible Immobilization (SPRI) bead kit (∼$1250), a 96-well magnet for the SPRI cleanups (∼$620), restriction enzymes (∼$2–300), a high accuracy DNA quantification kit (∼$100, plus ∼$2000 if an accurate fluorometer is not already available), plus the initial order of the custom oligo adapters with barcode sequences (which could also run into thousands of dollars depending on the number of enzymes and barcodes desired for the protocol). If all of those reagents are used up fully in several hundred library preparations, the cost per library will be quite low, on the order of $5–10/sample, whereas the cost of sequencing (∼$250 per sample) remains fixed. However, if only a few libraries are made before those reagents expire, the cost of the reagents alone may be greater than the total price for the 30 ezRAD libraries we ran here. Even using ezRAD, labs who plan to do only a few libraries are unlikely to want to invest in the Illumina TruSeq sample library preparation kit (∼$2600), and are much better off sending out to a commercial service for their needs at a higher price per sample, but much lower overall cost. Ultimately, there are trade-offs for every method, and for labs that plan to prepare and run many RADseq libraries, there are more cost-effective options available (Table 3); however for labs that need only a few runs for SNP discovery or marker development for targeted amplicon sequencing, and have none of the required supplies listed above in hand, ezRAD is the least technically challenging and most cost-effective option currently available.

Pooled versus unpooled libraries

In comparing pooled and unpooled libraries, we find that SNP discovery is more cost effective in libraries constructed with DNA from pooled individuals in comparison to multiple runs of single individuals (Fig. 1). For a substantial examination of pooled NGS sample strategies see Gautier et al. (2013). Here, our simple comparison of one library per individual on eight individuals of the reef fish Paracirrhites arcatus relative to two pools of four individuals each indicate pooled libraries are more cost-effective for SNP discovery (Fig. 1, File S3). Comparing our individual libraries illustrates that there is some variability among libraries in the markers recovered, likely due to imprecision in the gel size selection step; ultimately, the proportion of SNPs that were shared among all individuals drops as we compare across more of the individual libraries (File S3). Still, the highest coverage SNPs tend to be those shared among libraries, and the slope of loss asymptotes after about 5 individuals for reasonable levels of coverage (File S3). Even without normalization for lane use, we identified 3× more shared SNPs from the pooled libraries than from the individual libraries (Table 1, File S3). Although we used only eight individuals in this simple comparison of pooled and unpooled libraries, the best results for SNP discovery were in pooled libraries in which we had 20 or more individuals and other genomic resources against which to compare our variable base calls (e.g., P. miniata or C. talcosa, Table 1). Based on our findings here, our future efforts for SNP discovery would likely use two pooled libraries of many individuals (>20) each as the most cost-effective strategy to identify hundreds to thousands of high quality shared SNPs or loci for targeted amplicon sequencing (e.g., Puritz, Addison & Toonen, 2012) that could be reliably used for genotyping from these libraries (File S3).

Figure 1 Bar graph comparing pooled and unpooled libraries of Paracirrhites arcatus.

Relative proportion of high quality mapped reads, total SNPs, shared SNPs with greater than 10× coverage, and cost when employing one of two strategies: (1) preparing one library for every individual (8 individuals here), or (2) preparing two libraries of four pooled individuals. For all categories except Cost, taller bars represent better performance.

Benefits and trade-offs of ezRAD

Ultimately, there are tradeoffs to consider for each of the various approaches to reduced representation genomic sequencing strategies (reviewed by Wang et al., 2012). Although ezRAD is simple, works without optimization across a broad diversity of metazoan taxa, and requires little initial investment beyond the direct cost of NGS, it is important to consider that this approach, like many others, will not survey all restriction cut sites in the entire genome. Likewise, for labs in which a large number of libraries will be generated, another approach may prove more cost-effective (Table 3). However, for many population genomics applications, this limitation is unlikely to be of consequence, especially if the RAD sequencing is used to identify variable markers used in SNP analyses or for targeted amplicon sequencing (e.g., Puritz, Addison & Toonen, 2012). Our approach provides a greatly simplified and standardized approach that can be used to obtain RADseq libraries from a wide range of species, especially those for which little or no genomic information exists to guide restriction enzyme selection. Another potential concern often raised with reduced representation approaches is that of ascertainment bias – the systematic deviations from theoretical expectations that result from the sampling processes used to discover and measure their population-specific allele frequencies. The only RADseq study published for a marine invertebrate to date, Nematostella vectensis, has an available genome against which to compare results from the reduced representation genomic sequencing. While always a concern for such studies, Reitzel et al. (2013) tested for ascertainment bias among RAD loci using the available reference genome for N. vectensis and found no evidence of it for this species. They also compare results from their RADseq library with and without use of the reference genome to demonstrate that the lack of an available genome would have had no substantial impact on the results they obtained in terms of the number of SNP loci, detection of population genetic structuring or detection of loci under selection (Reitzel et al., 2013). Thus, particularly in the realm of marine invertebrates, for which species with available genomes are particularly sparse, RAD approaches hold considerable promise to access genomic information. The various methodologies of RADseq are rapidly becoming the standard approach for a wide range of studies, and ezRAD adds to the growing suite of methods available for sequencing reduced representation genomic libraries to access genomic information in non-model organisms.

Supplemental Information

File S1 Detailed protocol and modifications to the Illumina TruSeq Kit protocol for ezRAD

Click here for additional data file.

File S2 Bash script pipeline for processing ezRAD sequence reads

Click here for additional data file.

File S3 Comparison of pooled and unpooled libraries for SNP discovery in the reef fish Paracirrhites arcatus.

Click here for additional data file.

We are grateful to Amy Eggers of the Hawaiʻi Institute of Marine Biology EPSCoR Evolutionary Genetics Core Facility for her assistance in the preparation of the libraries and sequencing. We also thank Brian Bowen, Steve Karl, Mahdi Belcaid, Dan Barshis, Matt Iacchei, Mareike Sudek, and Ingrid Knapp, as well as all the other members of the ToBo lab for their discussion, advice and support of this work, and putting up with us taking over the lab to try out this crazy idea. We also thank Rick Grosberg and Brenda Cameron for generously allowing C. Bird to work on early RADseq attempts in their laboratory. This is contribution 1572 from the Hawaiʻi Institute of Marine Biology and SOEST 9032.

Additional Information and Declarations

Competing Interests

Author Contributions

Field Study Permissions

Data Deposition

Robert J. Toonen is an Academic Editor for PeerJ. None of the other co-authors have competing interests.

Robert J. Toonen conceived and designed the experiments, contributed reagents/materials/analysis tools, wrote the paper, conceived of the ezRAD approach.

Jonathan B. Puritz conceived and designed the experiments, performed the experiments, analyzed the data, contributed reagents/materials/analysis tools, wrote the paper, performed all the library preparations.

Zac H. Forsman and Jonathan L. Whitney analyzed the data, contributed reagents/materials/analysis tools, wrote the paper.

Iria Fernandez-Silva and Kimberly R. Andrews contributed reagents/materials/analysis tools, wrote the paper.

Christopher E. Bird conceived and designed the experiments, analyzed the data, contributed reagents/materials/analysis tools, wrote the paper, performed initial RADseq tests.

The following information was supplied relating to ethical approvals (i.e., approving body and any reference numbers):

No permissions needed for field study locations, and all samples from Hawaiʻi were collected legally under DAR special activities permits between 2008–2010 from the State of Hawaiʻi to the coauthors of this paper. Spinner dolphin samples were collected under NMFS Scientific Research Permits Nos 1007–1629 and 1000–1617 and University of Hawaiʻi Animal Care Committee Protocol No. 01–014 to K Andrews. Samples of sea stars were collected in 2008 and 2009 under California Fish and Game Permit SC-10066 issued to J Puritz.

The following information was supplied regarding the deposition of related data:

The following data can be requested from the following individuals:

Cellana talcosa: Chris Bird, chris.bird@tamucc.edu.

Cryptasterina hystera; C. pentagona; Patiria miniata: Jon Puritz, jpuritz@gmail.com.

Paracirrhites arcatus: Jon Whitney, jonathanlyonwhitney@gmail.com.

Pocillopora damicornis; Porites compressa; P. lobata: Zac Forsman, zforsman@gmail.com.

Stenella longirostris: Kim Andrews, kimandrews@gmail.com.

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
