# Peer review of "ezRAD: a simplified method for genomic genotyping in non-model organisms"

_PeerJ, doi:10.7717/peerj.203_

## Round 0.1 · original submission · Major Revisions

Both reviewers agree that this is an important contribution that needs a bit more work. They have provided useful comments to improve the manuscript. Please include a more thorough comparison of all RAD methods and how they differ with ezRAD. Also include more detail on the bioinformatics pipeline as highlighted by the reviews.
The concern raised with validation of the approach seems warranted and I encourage you to consider the suggestion of adding some experimental data if at all possible. It would strengthen the value of your method significantly.

Reviewer 1 ·

Basic reporting

The Authors present a variation of the Restriction-site Associated DNA (RAD) sequencing protocol for SNP discovery and genotyping in non-model organisms. By using MboI and Sau3AI, two restriction endonucleases with a very frequent recognition site (GATC), their approach directly produces fragments in the 300-400 bp range, thereby circumventing the sonication step of the original RAD protocol (Baird et al. 2008; Etter et al. 2011; Hohenlohe et al. 2010), which requires a specific and quite expensive piece of equipment. In addition, the protocol is compatible with the standard Illumina TruSeq library preparation kit, which provides the opportunity to outsource library preparation. The approach is interesting, the manuscript is generally clear and well-written. General comments are provided below, followed by specific comments.

General comments

The indexing and pooling strategies need to be clarified. By indexing I mean the ability to attribute each sequence to a specific individual using a specific ‘barcode’ or ‘index’ sequence tag. My understanding is that each library (not each individual, unless there is only one individual per library) is indexed, is that correct? The number of ‘tagged individuals’ in Table 1 varies between 1 and 8, but it is to be noted that several genotyping-by-sequencing (GBS) applications require the genotyping of tens to hundreds of indexed individuals. If my understanding of this approach is correct then one library would be required for each individual for such applications, which would be prohibitively expensive and time-consuming. A large part of the costs associated with RAD sequencing and similar approaches comes precisely from the number of different adapters (with different indexes) needed to index many individuals. If all is needed is a handful of individuals (or pools of individuals) then the classic RAD approach is not that expensive in terms of adapters/primers needed.

This brings me to my second comment. The Authors point out that thousands of SNPs can be identified and genotyped without optimization. But here again if all is needed is several thousand SNPs from a relatively small number of samples then the other RAD approaches do not require much optimization either. What requires optimization is the targeting of a specific number of loci (high or low), and a specific coverage per locus per individual in a specific number of samples. This kind of optimization requires the analysis of a preliminary dataset, regardless of the approach taken. More generally, what is and what is not different from other GBS approaches should be clarified. Other GBS approaches do not require a genome as well, and some of them (e.g. double-digest) do not require a sonication step either. Other GBS approaches also have a size selection step which makes them scalable to a certain extent. Also the RAD approach is not specific to rare cutters, it can be used with frequent cutters as well. In many cases the same RAD adapters can be used with frequent and rare cutters. For example the adapters with TGCA overhang can be used with SbfI, PstI, or NsiI. For a 1 Gb genome with 41% GC content the expected number of fragments would range between about 13,000 (with SbfI) and 640,000 (with NsiI). I acknowledge that MboI and Sau3AI have more frequent recognition sites than the endonucleases typically used with the classic RAD protocol, but this could be clarified in the manuscript.

Figure 1 illustrates a repeatability issue in terms of SNPs shared between different libraries. Gel size-selection is a cheap and accessible but very imprecise procedure, regardless of how cautiously it is done. As a result different libraries tend to have a large proportion of fragments that are not shared, which significantly limits the number of markers that can be used in the end (unless, here again, many indexed individuals are pooled in the same library). The use of a very frequent cutter is expected to exacerbate this issue, because there are so many fragments. One option would be to size-select more precisely, e.g. using the BluePippin platform by Sage Science, but as for the sonicator this is a quite specific and expensive piece of equipment. This issue should be addressed in the manuscript.

Generally speaking more details are needed on the analysis pipeline (the script by itself is useful but cryptic to most readers). How are the reads assembled into contigs? What parameters are used? How sensitive is the method to sequencing errors and duplications? This should be drastically expanded in the Methods, Results, and Discussion.

Specific comments

It is mentioned in the Methods that BWA is used to ‘map the reads to the assembly’, what assembly is that? This approach is presented for non-model organisms, so I am assuming that there is no genome (assembly) available for the species considered is that correct? I also note that in the absence of a genome a higher coverage (in the 30x-60x range) is advisable.

I finally note that the observations that i. the results are highly dependent on initial DNA amount and quality and ii. that pooling individuals is cheaper are both expected for all GBS approaches.

Experimental design

See 'Basic Reporting'

Validity of the findings

See 'Basic Reporting'

Additional comments

See 'Basic Reporting'

Reviewer 2 ·

Basic reporting

Toonen and coauthors present a variation on RAD library preparation and demonstrate its application to a selection of non-model marine species. I welcome this RAD variant to the growing family of methods used for genotyping by sequencing reduced representation libraries, and consider the study deserving of publication in principle. There are many commendable aspects of this manuscript. I appreciate the authors’ goal of demonstrating application of RAD genotyping in non-model species, especially marine invertebrates, which they rightly observe has lagged behind other systems. And I appreciate the authors’ inclusion of commented scripts for bioinformatics analysis, and detailed protocols for library preparation.
However, several issues remain to be addressed before I can recommend the manuscript for publication. I have organized my suggestions into “major revisions”, which I view as required for publication, and “minor revisions” which I view as optional but hope the authors will consider adopting.
Major revisions
1. First, I don’t think the comparison of existing RAD methodologies is adequate. In my view, each of the extant methods offers advantages and disadvantages, but many of these have been glossed over in the discussion. Advantages unique to extant methods that are not shared by ezRAD are not mentioned. For example, the sonication employed in standard RAD, in combination with PE sequencing, allows for de novo assembly of long contigs that may be useful for many applications. Both 2bRAD and ddRAD offer methods for reducing marker density and sequencing cost that do not seem to be available in ezRAD. I suggest the authors rewrite this section extensively in an effort to present an objective comparison of the advantages and disadvantages of each method.
In addition to omitting some important details, the authors suggest that the strategy of restricting sequenced fragments by combining restriction digest with size selection is novel: “ezRAD, a novel approach to reduced representation genomic sequencing, differs from existing RADseq methods in that we employ a single frequent recognition site (GATC) and then limit library inclusion based on size selection of a portion of the digested fragments”. In fact, this strategy is the same as ddRAD, although different enzymes are used. As I understand it, the only thing that is truly novel about ezRAD is that it uses Illumina TruSeq kits for all library preparation steps downstream of the restriction digest. This may be viewed as an advantage by some, since it means less expertise in library preparation is required. However, the disadvantage is that these kits are an order of magnitude more expensive than the “homemade” library preparations in other RAD variants. The authors have included estimates for some components of the cost, but it would be useful to compare both the library preparation and sequencing components of various methods. This will highlight the extremely high cost associated with using TruSeq kits (~$300 per sample) relative to homemade preparations ($5-$10 for 2bRAD or ddRAD).
2. The authors chose a coverage threshold (5X) much lower than typically used for RAD analysis (generally 20-30X). This concerns me because it is difficult to tell the difference between sequencing error and heterozygous genotypes at such low coverage, or, indeed, between heterozygous and homozygous genotypes. Consider the cumulative binomial probability function. At 5X coverage, the minor allele will be detected less than twice for 18% of heterozygous loci, even if we generously assume that both alleles are present in the library at exactly 50%. At 20X coverage, this probability falls dramatically to 2e-5. Please discuss coverage threshold and justify your choice of such a low threshold. Of course, with realistic deviations from the 50:50 ratio during library preparations these error rates will become both higher and more difficult to estimate.
3. On a related note, I strongly recommend some experimental validation of these genotype data. Cursory examination of any NGS dataset reveals a large number of sequence variants; distinguishing between artifacts (sequencing errors or errors introduced during library preparation) and biologically true SNPs is the central challenge for all methods for genotyping by sequencing. In the current manuscript, all sequence variants are referred to as SNPs. At minimum, this issue needs to be discussed. Ideally, I would like to see experimental validation. Simply designing flanking primers and Sanger sequencing amplicons from the same DNA samples initially used for library preparation for ~20 loci would accomplish this goal for a modest investment of time and resources.
Further, a common framework for “methods” publications like the present study is to describe the method, then demonstrate its utility in answering a biological question. The current version of the manuscript falls short in the latter respect; a critical reader may notice that all the authors have truly demonstrated is that ezRAD is capable of producing sequence data for many species, which I think we can all agree is a pretty low bar. Demonstrating the accuracy of these genotypes would greatly strengthen the manuscript.
Minor revisions
1. Please convert all C-values to megabases. In the age of NGS, units of base pairs are more relevant; I expect that all your readers will be mentally performing this conversion anyway so its better to save your readers the effort.
2. “For example, Acropora digitifera has ~420 megabase genome (Shinzato et al. 2011), and in combination with the suite of dinoflagellate, prokaryotic and eukaryotic symbionts inextricably associated with the coral holobiont (reviewed by Ainsworth et al. 2010) this makes for an exceedingly large genomic pool from which to draw fragments for reduced representation genomic sequencing.”
An important caveat for this statement is that, assuming standard DNA extraction methods, the genomic DNA extracted from such a sample will be almost entirely derived from the animal host so that the microbial and algal contributions are negligible, from the perspective of the tags genotyped (these contaminants will typically fall below any reasonable coverage threshold).
3. Please archive all sequences in a public database (NCBI’s SRA would be appropriate) and provide accession numbers so that interested readers can reanalyze these data using other software (e.g. the widely used STACKS).
4. The manuscript would benefit from expanding the Bioinformatics section of Methods. Rather than simply listing a software package, it would be ideal to very briefly explain what that package accomplishes. Many options exist for SNP calling algorithms in particular and it would be useful to explain how genotypes were inferred from nucleotide frequencies in this study. Also, please be sure to state any non-default settings or arguments used in each package. In a methods paper such as this, I encourage the authors not to skimp on the details of bioinformatic analysis.
5. The manuscript consistently frames RAD sequencing in the context of SNP discovery, rather than SNP genotyping itself. However, for projects that require relatively large numbers of markers (100s or greater), and projects focusing species with small research communities, it is typically more cost effective to use RAD for genotyping itself rather than developing targeted SNP assays. Please discuss this issue in your expanded comparison of RAD methods. When genotyping by RAD, one challenge frequently encountered is missing data: some loci are genotyped in one sample but not another. The present study genotyped two samples of Paracirrhites arcatus that present an opportunity to explore this question. What fraction of the loci genotyped in one sample were also genotyped in the other?
6. Table 1. Several genus names are broken across lines.

Experimental design

No comments -- all details covered in "Basic Reporting" section.

Validity of the findings

No comments -- all details covered in "Basic Reporting" section.

---

## Round 0.2 · accepted · Accept

I appreciate your thorough and thoughtful responses to the feedback from the reviewers. I consider this an important contribution to a fast-evolving methodology. The approach will certainly provide options to the community. I am glad this manuscript will be published in PeerJ.